# mRNA: Vaccine or Gene Therapy? The Safety Regulatory Issues

**DOI:** 10.3390/ijms241310514

**Published:** 2023-06-22

**Authors:** Helene Banoun

**Affiliations:** Independent Researcher, 13001 Marseille, France; helene.banoun@laposte.net; Tel.: +33-6-32-46-78-33

**Keywords:** mRNA vaccines, COVID-19, drug regulation, gene therapy, vaccinovigilance, pharmacokinetics

## Abstract

COVID-19 vaccines were developed and approved rapidly in response to the urgency created by the pandemic. No specific regulations existed at the time they were marketed. The regulatory agencies therefore adapted them as a matter of urgency. Now that the pandemic emergency has passed, it is time to consider the safety issues associated with this rapid approval. The mode of action of COVID-19 mRNA vaccines should classify them as gene therapy products (GTPs), but they have been excluded by regulatory agencies. Some of the tests they have undergone as vaccines have produced non-compliant results in terms of purity, quality and batch homogeneity. The wide and persistent biodistribution of mRNAs and their protein products, incompletely studied due to their classification as vaccines, raises safety issues. Post-marketing studies have shown that mRNA passes into breast milk and could have adverse effects on breast-fed babies. Long-term expression, integration into the genome, transmission to the germline, passage into sperm, embryo/fetal and perinatal toxicity, genotoxicity and tumorigenicity should be studied in light of the adverse events reported in pharmacovigilance databases. The potential horizontal transmission (i.e., shedding) should also have been assessed. In-depth vaccinovigilance should be carried out. We would expect these controls to be required for future mRNA vaccines developed outside the context of a pandemic.

## 1. Introduction

The regulation of medicines and vaccines is a little-known but very important subject. Indeed, health products must undergo very strict controls, as a principle, in order to control their efficacy and safety profile.

The anti-COVID-19 mRNA vaccines are the first mRNA vaccines marketed. mRNA vaccines, which represent a new class of vaccine, should be subject to more controls than conventional vaccines because they are based on several new technologies [1]. Although incompletely defined, the mode of action of mRNA vaccines [2] should classify them as gene therapy products (GTP) [3]. But mRNAs as vaccines against an infectious disease have been excluded from GTP regulation by US and EU regulations [4]. No specific regulations existed before the year 2020 for mRNA vaccines. “The current guidelines either do not apply, do not mention RNA therapeutics, or do not have widely accepted definition” [5]. Regulatory agencies therefore had to adopt an emergency procedure to monitor the testing of these products, the rolling review. In rolling reviews, data are submitted and reviewed as they become available before the full data package is available and specific controls for this new platform have been requested [6].

The aim of this study is to compare the controls required by GTP regulations with those actually applied to mRNA COVID-19 vaccines. Some of the controls required for GTPs were not required for mRNA COVID-19 vaccines, probably because of the pandemic emergency that required the rapid development of SARS-CoV-2 vaccines. Potential safety issues arising from the absence of these controls will be discussed. This is all the more urgent as manufacturers are planning to replace certain “classic” vaccines with mRNA vaccines [2], starting with influenza vaccines. Indeed, Sanofi is launching a clinical trial of the first mRNA-based seasonal flu vaccine candidate [7] and Moderna has many mRNA vaccines in clinical trials (COVID-19, influenza, human metapneumovirus, parainfluenzas, RSV, HCoV, CMV, EBV, HSV, varicella, herpes, HIV, Zika, Nipah), in particular a phase 3 trial of the flu vaccine [8].

A phase 1 clinical trial is being launched for an mRNA-LNP influenza vaccine [9]. For these flu vaccines, emergency approval should not apply and the requirement for these additional studies should not be exceeded.

In addition, cancer “vaccines” are being announced (e.g., Moderna and Merck are partnering in trials of mRNA-4157/V940, an anti-melanoma “vaccine” combined with Keytruda—a monoclonal antibody directed against the programmed cell death receptor, PD-1) that acts by enhancing the ability of the body’s immune system to detect and fight tumor cells, by blocking the interaction between PD-1 and its ligands, PD-L1 and PD-L2, thereby activating the anti-T cell response, particularly the antitumor response [10]).

We must be very vigilant about the term vaccine associated with therapeutic drugs, particularly with regard to the regulations that apply to them. These therapeutics are not vaccines against infectious diseases and must therefore continue to comply with GTP regulations.

## 2. Current Regulation of Anti-COVID-19 mRNAs

### 2.1. The Mode of Action of mRNA Anti-COVID-19 Defines Them as GTP and Their Destiny as Vaccines

Although incompletely defined, the mode of action of mRNA vaccines should classify them as gene therapy products (GTP) [2]; they are nucleic acids intended to make the cells of the vaccinee produce an antigen, inducing the production of antibodies. This mode of action corresponds exactly to the regulatory agencies’ definition of a GTP.

According to the EMA CHMP report (Committee for Medicinal Products for Human Use) [11]: “The active substance consists of mRNA that is translated into the spike protein antigen of SARS-CoV-2; the LNP protects the RNA and enable transfection of hosts cells after IM delivery; the S protein induces an adaptive immune response”.

According to the FDA [12], gene therapy is a medical intervention based on the modification of the genetic material of living cells. Cells may be altered in vivo by gene therapy given directly to the subject.

According to EMA 2009 [13], a GTP: (a) contains an active substance which contains or consists of a recombinant nucleic acid used in or administered to human beings with a view to regulating, repairing, replacing, adding or deleting a genetic sequence; and (b) in its therapeutic, prophylactic or diagnostic effects, relates directly to the recombinant nucleic acid sequence it contains, or to the product of the genetic expression of this sequence.

But mRNA also corresponds to the definition of a vaccine; there is, however, an incompatibility to a certain point.

The 2005 WHO guidelines [14] grant nucleic-acid-based vaccines the status of vaccines (“*antigens produced in vivo in the vaccinated host following administration of a live vector or nucleic acid or antigens produced by chemical synthesis in vitro*”).

According to the definition of the ANSM (Agence Nationale de Sécurité du Médicament, Saint-Denis, France) [15], a vaccine is a preventive medicine composed of one or more active substances of biological origin called antigens administered to protect against a disease.

According to the CDC [16], a vaccine is a preparation that is used to stimulate the body’s immune response against diseases.

According to European regulations, vaccines are products capable of producing active immunity [17] and contain antigens capable of inducing active immunity against an infectious agent [4]. According to the EMA [11], the active substance of the COVID-19 Pfizer vaccine is mRNA: it is not an antigen. Therefore, according to the European and French pharmacopoeias, mRNAs should not be considered as vaccines because they do not contain antigens.

Moreover, still according to their mode of action, mRNA vaccines can be considered as pro-vaccines; this is a neologism modelled on the word pro-drug which designates a drug which, after administration, is converted by the organism into a pharmacologically active drug. In fact, according to the principle of mRNA, this must be translated into protein by the cells of the person vaccinated (the injected substance is not the substance causing an active immunization). According to the FDA [18], mRNA vaccines correspond to the TypeIA of pro-drugs, which are substances that are converted by cells into active drugs. This pro-drug property could imply additional controls to those applied to vaccines. However, neither the FDA nor the EMA make any reference to these qualifications for mRNA anti-COVID-19 vaccines.

### 2.2. mRNAs as Vaccines against Infectious Disease Have Been Excluded from GTP Regulation by US and EU Regulations

In 1998, the FDA already stated that recombinant DNA materials used to transfer genetic material used as preventive vaccines are not covered by the guidance for gene therapy [12]. In 2007, the FDA distinguished between DNA plasmid vaccines according to whether they were for the prophylaxis of an infectious disease or not; DNA plasmids without indication in infectious diseases were subject to the regulation of gene therapy products, whilst DNA plasmid vaccines against infectious diseases were subject to a separate regulation inspired by those for GTP [19]. In 2013, the FDA confirmed that the regulation of gene therapy products did not apply to vaccines against infectious diseases [20]. In a 1996 document [21] on DNA plasmids for preventive infectious disease indications, it is specified (as for all FDA guidelines) that no subsequent requirements were established: these were non-binding recommendations. For this reason, this study will be based primarily on EMA documents; although EMA guidelines are not legally binding, applicants need to provide justification for any deviations.

According to the EMA, since 2009, “Gene therapy medicinal products shall not include vaccines against infectious diseases” [13]. This exclusion was confirmed in 2015 [22]. For a history of the regulation of nucleic acids to prevent infectious diseases, see Table 1.

This exclusion poses a logical problem because not all RNA-based products have the same regulatory status, as Guerriaud and Kohli have pointed out [4].

### 2.3. It Is Necessary to Underline the Contradictions of the Legislation

According to European Union (EU) legislation, RNA-based medicines can currently be classified into different regulatory statuses, depending, for vaccines, on their target (infectious disease or not) and, for other medicines, on the way they are obtained (chemically or biologically) [4]. This classification determines the controls and studies that must be carried out to obtain marketing authorizations. Thus, mRNA vaccines against infectious diseases are not classified as gene therapy products [17] (p. 162) and [22] whereas mRNA vaccines for the treatment of cancers are GTMPs (gene therapy medicinal products, which are part of ATMPs, advanced therapeutic medicinal products); in fact, mRNAs are GTMPs, according to the CAT (Committee for Advanced Therapies), and must therefore undergo complete pharmacokinetic studies [24].

It is therefore surprising that Moderna and BioNTech expected to have their products regulated as gene therapies. Moderna, Inc. acknowledged in its Q2 2020 Securities and Exchange Commission (SEC) filing that “*currently, mRNA is considered a gene therapy product by the FDA*” [26]. Furthermore, BioNTech founder, Ugur Sahin, in a 2014 article, stated “*One would expect the classification of an mRNA drug to be a biologic, gene therapy, or somatic cell therapy*” [27]. Thus, the status of COVID-19 mRNA vaccines was not well understood by the manufacturers themselves.

The EMA regulation must refer to the WHO guideline on the nonclinical evaluation of vaccines. According to the EMA “In contrast to the EU guideline, the WHO guideline also includes vaccines containing a live vector or nucleic acid. This is considered appropriate since many of the nonclinical aspects to be addressed are common for traditional vaccines and vaccines containing a live vector or free DNA”. This is in contradiction with the exclusion of nucleic acid vaccines from the GTP regulation, as the EMA specifies in the same 2016 document that the specific aspects of nucleic acid vaccines must be studied in light of GTP regulations [28].

In 2020, the WHO noted the lack of clarity in the regulations governing mRNA vaccines. To solve this regulatory problem, the WHO published a draft guidance document on 20 December 2020, for the assessment of the quality, safety and efficacy of the mRNA vaccines, which includes the manufacture and control of vaccines as well as their non-clinical evaluation. The WHO admitted that detailed information was not available for the production of COVID-19 mRNA vaccines. In addition, the safety and efficacy controls for gene-based biologics were not standardized, and some details remained proprietary and were not publicly disclosed. Given these uncertainties, the WHO felt that it was not possible to develop specific international guidelines or recommendations and that some regulatory flexibility was needed [25].

### 2.4. Why Are mRNA Vaccines Excluded from the Regulation of Gene Products?

According to Guerriaud and Kohli [4], “*it is difficult to answer with certainty why vaccines against infectious diseases have been excluded. The definition [of vaccines] has not changed since 1975, a period when there was no “vaccine” against cancer*” [17]; they are agents capable of producing active immunity against an infectious disease. At that time, the only existing vaccines were against infectious diseases and the current definition of a vaccine is limited to an immunological drug against an infectious disease. Therefore, an anti-cancer drug can, in no way, be called a “vaccine”. It should be noted that therapeutic AIDS vaccines, based on lentiviruses and acting as gene therapies, because they integrate into the genome, have also been excluded from gene therapies [29]. It can be assumed that the applicant has argued that the product has both a therapeutic and a prophylactic mode of action, but the document is not available on the EMA website.

From a public health point of view, and knowing that anti-COVID-19 mRNAs considered as vaccines have not undergone all the strict controls required for GTPs (see below), one could object that a product intended for the majority of the world’s healthy population should be subject to more stringent regulation than a GTP intended for a few rare people suffering from a rare disease or cancer (this time concerning millions of people). Moreover, according to the EMA [28], “*Since vaccines in most cases are given to large numbers of healthy individuals, there is a need for a solid nonclinical safety evaluation*”.

This exclusion could have a regulatory explanation; this decision was made partly because vaccines have a very different mechanism of action than other medicinal products and partly to ensure that all vaccines are reviewed by the same committee (The Committee for Medicinal Products for Human Use—CHMP), the special vaccine experts of the EMA (The Scientific Advisory Group on Vaccines (SAG-V) [30]) and the Vaccines Working Party [31].

### 2.5. Required Controls for mRNAs Considered as Vaccines

Vaccines, in general, are part of human medicines according to the EMA and must therefore undergo the controls pertaining to these products; the regulation of a drug concerns good manufacturing practices (GMPs). These GMPs are detailed in the EMA document of 2001, updated in 2012, which applies to all human drugs including vaccines [17] (p. 117). These GMPs concern, among other things and for preclinical studies the assessment of environmental risks, the characterization of the product and raw materials (purity and quality), their control and stability, manufacturing methods, pharmacology studies (e.g., quantitative composition, description of the manufacturing process, raw materials not listed in a pharmacopoeia, identification and assay of the active substance, mode of action, in vitro or in vivo test of biological activity—if assaying in the finished product is not possible—toxicity, carcinogenicity, reproductive and embryo/fetal toxicity, pharmacokinetics, pharmacodynamics—modification of physiology by the drug—efficacy and product safety). For a new excipient, the chemical, pharmaceutical and biological information should be identical to that provided for the active substance [17] (p. 135). The WHO recommends the same type of controls for vaccines in general [14].

GMPs concerning product quality, purity, stability, manufacturing methods, reproductive and embryo/fetal toxicity and carcinogenicity will be discussed below. Although these GMPs apply to all human medicines, they are not generally applied to vaccines.

Pharmacokinetics is the action of the organism on a drug, i.e., the fate of the drug, from its entry to its exit from the organism, the evolution over time of its absorption, its bioavailability, its distribution, its metabolism and its excretion [17]. “*Pharmacokinetic studies are usually not required for vaccines. However, such studies might be applicable when new delivery systems are employed or when the vaccine contains novel adjuvants or excipients*” [23].

According to a 2016 document [28], for the EMA, the regulations should follow those of the WHO. The WHO guidelines [14] specified that a pharmacodynamic study may also extend to the pharmacology of an adjuvant and that distribution studies must be considered in the case of new formulations. When a new additive is to be used, for which toxicological data are not available, toxicity studies of the additive alone should first be performed and the results documented in accordance with the guidelines for new chemical entities.

The EMA also requires additional studies for vaccines using a new formulation; a 2006 document, applicable to “DNA vaccines expressing foreign antigens”, states that “Pharmacokinetic studies are usually not required for vaccines. However, such studies might be applicable when new delivery systems are employed or when the vaccine contains novel adjuvants or excipients” [23]. This document was drawn up before the exclusion of nucleic acid vaccines against infectious diseases from the GTP regulations. We will see below that the pharmacokinetic studies provided for anti-COVID-19 mRNAs are incomplete.

### 2.6. Controls Required as a Pro-Drug

The FDA [18] points out the particular problems of pro-drugs raised by their more or less complete conversion into an active substance and the question of toxicity. According to the FDA, it is necessary to define how the pro-drug contributes significantly to the toxicity profile of the active drug, particularly as a function of the site of transformation and action. For mRNA vaccines, biological transformation occurs in many cell types and in all organs (see below), whereas the desired goal, i.e., immunization, will only occur in immune cells. However, mRNA vaccines are not classified as pro-drugs and therefore do not have to undergo the controls concerning the site of transformation and action.

### 2.7. Additional Controls Based on GTP Regulations

Given the pandemic emergency, the vaccine review process has been modified and accelerated in the form of a rolling review. In rolling reviews, data are submitted and reviewed as they become available, before the full data package is available. This approach requires a closer collaboration and more intense interaction between the sponsor and the health authority [6]. It can be inferred from the EMA reports that the EMA has adapted the vaccine regulations and required certain controls specific to gene therapy products. This is what emerges from the analysis of the EPARs for Moderna [32] and the rolling review for Pfizer [11,33,34].

Despite the absence of specific regulations for mRNA vaccines, the EMA has added the following controls on product quality: identity (by RT-Sanger sequencing for Spikevax and Next Generation sequencing for Comirnaty); total RNA content (UV), purity (RP-HPLC); product-related impurities (RP-HPLC); % 5′ Capped (RP-UPLC); % PolyA tailed RNA (RP-HPLC for Spikevax, not fully described for Comirnaty) and residual DNA template (qPCR).

However, some specific obligations reported in the first report (for example, the mode of action, which is not described) [11,33,34] have not been fulfilled according to the 2021 report [35]. Regarding impurities, the method of determination of bacterial endotoxin should be specified [11,33,34]. The levels of endotoxin found are not specified in the documents of the Australian regulatory authority, the TGA, and some batches were still under evaluation at the time of batch release [36]. The accepted limit is 12.5 EU/mL [34].

Concerning the quality of the product, the in vitro transcription method is not sufficiently described and the characterization of the mRNA is not satisfactory; the methods of evaluation of truncated and modified mRNAs must be more detailed. A potency test is not satisfactory, in vitro expression test needs to be updated. If the Poly(A) tail length and percentage partly remains, the REC20 is not fulfilled. If 3 months of stability data are provided, 6 months of data are expected [36], and the presence of truncated mRNA and truncated protein expression is not sufficiently explored [33]. EMA calls for a clarification of the mode of action [11,33,34].

Therefore, the results of the controls required for GTPs that were requested by the EMA are not sufficient according to the EMA reports.

## 3. Controls Required by GTP Regulations to Which Anti-COVID-19 mRNAs Were Not Subjected

### 3.1. Product Quality

Concerning product quality, GTPs are subject to specific controls not mentioned for non-gene drugs [37,38]. Among these controls, the endotoxin level was not numerically provided (see above) and the interaction of nucleic acids with the vector was not studied.

The presence/absence of specific features, such as CpG sequences, should be confirmed by suitable methods: this is not provided. The research and quantification of product-related impurities (deleted, rearranged, hybrid or mutated sequences, oxidation, depolymerisation) are not provided. The use of antibiotic resistance genes in final GTMP should be avoided if possible and, where not possible, justified (this was not justified). “If unavoidable, studies should be performed before first clinical studies addressing inadvertent expression of the resistance gene in human somatic cells” [37]; these studies have not been carried out.

Concerning the US-FDA, one should refer to the CBER (Center for Biologics Evaluation and Research) guide in charge of regulating these products, which only issues non-binding recommendations [39], as well as to the 2013 instructions [20], which, globally, impose the same criteria as the EMA.

### 3.2. Pharmacokinetics

According to the EU, GTMPs require specific tests or trials to evaluate the risk of genome integration and germ-line transmission [24,38], even if this integration is unlikely [37], as it is the case for RNA.

GTMPs require specific tests or trials to evaluate the risk of insertional mutagenesis, tumorigenicity, embryo/fetal and perinatal toxicity and long term expression [38] that have not been performed.

For GTMPs, the EMA requires extensive studies on both the nucleic acid and the vector particle/delivery system that includes biodistribution, dose study, potential target toxicity, the identification of the target organ to obtain biological activity, toxicity linked to the expression of structurally altered proteins, reproductive toxicity (for these studies, tests must follow the ICH M3 document [40]), repeated toxicity and excretion in the environment. These studies are required for products containing DNA because the document was drafted in 2006 [37] and mRNA vaccines were not considered at the time. Repeated toxicity was not adequately studied because only two doses of vaccine were planned [32].

It is necessary to insist on pharmacokinetic studies, which are generally not required for vaccines unless they are based on a new formulation or when the vaccine contains novel adjuvants or excipients. The need for such studies must be assessed for vaccines on a case-by-case basis by the regulatory authorities [23]. Moreover, “the standard absorption/distribution/metabolism and excretion studies for conventional medicinal products may not be relevant for GTMPs” [38]. For example, the route of administration that is considered as the worst case scenario (e.g., intravenous, representing the effect of widespread dissemination of the GTMP) should be considered. For GTPs, shedding studies are expected that address excretion and dissemination in the body, including studies on persistence, clearance and mobilization. Biodistribution studies should also address pharmacokinetic studies of the transgenic product (e.g., expressed proteins). The studies provided by the manufacturers seem to be incomplete from this point of view (see Section 4).

### 3.3. Controls on Biological Drugs Not Carried Out

The EMA, like the European Commission, considers that “RNA-derived products should be considered as biological products, even if they are not derived from a biological source” [41]. According to European regulations [17], for a biological drug, a list of biological activities must be provided, and studies of reproductive function, embryo-fetal and perinatal toxicity and mutagenic and carcinogenic potential must be considered. We have seen above that these tests have not been carried out, and that the biological activities of the active ingredient—mRNA—have not been sufficiently described.

### 3.4. Clinical Studies

The shedding studies (through secretions and excretion) in animal models will be used to estimate the likelihood and extent of shedding in humans and to guide the design of clinical shedding studies. Clinical pharmacokinetic studies should be included if the gene product is a protein excreted in the blood circulation. The potential for transmission to third parties needs to be investigated, or a justification for not doing this should be provided. The dose response effect should be evaluated. Shedding in the seminal fluid must also be addressed for GTP; there is no mention of this possible excretion in regulatory agency reports.

Genotoxicity issues, including insertional mutagenesis and consequent tumorigenicity, should be evaluated carefully in relevant in vitro/in vivo models. Immune suppression, a causative factor for tumorigenesis in humans, must be investigated. According to Spikevax-EPAR, no carcinogenicity, insertional mutagenensis nor tumorigenicity in in vivo studies were submitted. Embryo-foetal and perinatal toxicity studies may be required if women of child-bearing potential are to be exposed to GTMPs [38].

### 3.5. Vaccinovigilance

According to the EMA, GTMPs have an obligation to provide safety and efficacy data for 30 years after the expiry date of the drug, which is beyond the requirements of classical pharmacovigilance [24].

According to the FDA regulations for GTPs [42], a long-term follow-up of adverse events must be performed for at least 5 years for new clinical conditions, such as: new malignancy(ies), new incidence or exacerbation of a pre-existing neurologic disorders, new incidence or exacerbation of a prior rheumatologic or other autoimmune disorder, new incidence of a hematologic disorder and new incidence of infection (potentially product-related).

According to the European regulation [13], a strategy for the long-term follow-up of safety and efficacy shall be included in the risk management plan.

With regard to conventional vaccines, the duration of observation of adverse events is generally only a few weeks. The Brighton Collaboration [43], which is responsible for monitoring the safety profiles and benefit/risk ratios of vaccines, has published a guide for monitoring the selected adverse events of vaccines in general. Follow-up times are sometimes specified but rarely exceed 2 months [44] and, according to an FDA COVID-19 vaccine pharmacovigilance study [45], vaccinated individuals are followed for up to 42 days.

## 4. Discussion

### 4.1. Controls Required for a Pro-Drug That Have Not Been Carried Out

If anti-COVID-19 mRNAs had been classified as pro-drugs, they would have had to undergo controls concerning the site of transformation and action [18]. It would thus have been detected that the spike protein translated from the mRNA is not only found in the immune cells of the muscle where the mRNA is injected. This point will be discussed below in Section 4.3.1.

### 4.2. The Results of Certain Tests Required for Vaccines in General Are Not Satisfactory

#### 4.2.1. Drug Substance Purity

This is not the place to discuss the results of mRNA controls, but it seems important to do so solely with regard to product purity. The EMA requires a purity of 95% for products for human use [17]; according to EMA [11], the purity of the Pfizer final product is variable depending on the manufacturing process. According to the “Rapporteur Rolling Review critical assessment report” obtained by FOIA [34] (pp. 81 and 102), which details the previous document, the purity of the product is well below 95% at the time of marketing and the acceptance criterion is 50%. In another document obtained by FOIA, this threshold is 58% for mRNAs [35] (p. 38). In the 2022 report for the Moderna vaccine adapted to the Omicron strain, the EMA again asks to reassess the need to adjust the purity specification limits at the level of the active substance [46]. These defects in product purity are questionable for a new formulation.

It should be noted that the batch release procedure conducted by the OCABR [47] did not detect any batch heterogeneity. The document provided by the OCABR does not specify the controls performed by the reference laboratories regarding the identity, potency and integrity of the product. However, a heterogeneity in the toxicity of the batches has been published which could result from a heterogeneity of composition [48].

#### 4.2.2. Drug Substance Impurities

The specification for a residual DNA template was based on the WHO recommendation: no more than a 10 ng DNA/dose [49]. This limit had been set in 1985 by the FDA at 10 pg per dose of vaccine, and raised by the WHO in 1986 to 100 pg per dose, then 10 ng per dose in 1996.

The WHO pointed out that the total number of doses to be given should be taken into account when setting this limit. Based on these considerations, and assuming a maximum dose of 30 μg, the commercial acceptance criterion at release is ≤330 ng DNA/mg RNA [33] (p. 103). However, the EMA requests further information on the linear DNA template and the quantification method.

In the EMA report [33], the results for residual DNA template and ds-RNA assays were highly heterogeneous between batches, although well below the accepted limits. It would be wise to re-evaluate assay methods and limits for future mRNA vaccines that will be evaluated outside a pandemic period. This is all the more true given that the final number of doses of COVID-19 mRNA vaccine that an individual will receive is not yet known.

#### 4.2.3. Problem Posed by the Presence of Antibiotic Resistance Genes

The DNA plasmid used as a template for mRNA production contains a kanamycin resistance gene [33] (p. 26). Given the significant and variable quantities of contaminating DNA in the drug substance, there is concern that the resistance gene could be integrated into human digestive tract bacteria or somatic cells [37]. If anti-COVID-19 mRNAs had been subject to GTP regulation, these studies would have been carried out.

Therefore, the controls required for all drugs and vaccines have not given completely satisfactory results concerning the product purity and quality.

### 4.3. Controls Required for GTP That Were Not Performed: Safety Issues Arising from mRNA Pharmacokinetics

#### 4.3.1. Pharmacokinetics of Anti-COVID-19 mRNAs

The pharmacokinetic controls required for a new vaccine formulation have not been fully performed. It is unfortunate that complete pharmacokinetics studies have not been fully conducted since the EMA points out that several reports in the literature indicate that LNP-formulated RNAs can distribute rather non-specifically to several organs such as the spleen, heart, kidney, lungs and brain [11] (p. 54). Moreover, independent post-marketing studies have shown the distribution and persistence of the mRNA for several weeks in many organs [50,51,52]. The product of the mRNA, the spike, also circulates in the blood for several weeks [53,54,55,56]. The spike protein was found in the brain and the heart of a person who died 3 weeks after vaccination [57]. The spike was found in skin lesions up to 100 days after vaccination [58]. The preclinical studies provided by the manufacturers, therefore, appear to be incomplete from a pharmacokinetic standpoint, as they failed to detect this broad biodistribution and persistence.

For Pfizer, only the isolated components of the nanoparticles were studied regarding biodistribution. The EMA notes, from studies on similar components carried out for a GTP (the Patisiran data), that a half-life of 20–30 days in humans, and of 4–5 months for the complete elimination of lipids from nanoparticles, can be expected. Biodistribution should have been studied on the complete nanoparticle loaded with mRNA, all the more so as preclinical studies have shown biodistribution in all organs [32].

According to Spikevax-EPAR [11], biodistribution, genotoxicity and repeat toxicity studies were performed with mRNAs encoding proteins other than the SARS-CoV-2 spike. This is not compatible with the GTP regulations, as the EMA requires that distribution studies be conducted on the transgene, as included in the GTMP [37].

These biodistribution data should have reinforced the need for certain essential GTP controls. Indeed, the EMA [38] requires that, in the event of signs of long term expression, unintended genomic integration and oncogenesis must be investigated. The duration and expression should be determined by RT-PCR and immunological assays and/or assays to detect functional protein. The over-expression of the transgene has to be monitored [38]. This should have been controlled, given that large quantities of the spike protein can be produced, sometimes in excess of those circulating in those with severe COVID-19. A comparison of spike concentrations achieved during disease and after vaccination shows that, during severe COVID-19, the median concentration observed is 50 pg/mL, with maximums at 1 ng/mL. During severe COVID-19 infection, levels of up to 135 pg/mL of S1 spike can be detected, most commonly between 6 and 50 pg/mL. After vaccination with mRNA vaccine concentrations, up to 150 pg/mL are commonly observed, but may reach 10 ng/mL in individuals with vaccine-induced thrombocytopenia [55,59].

#### 4.3.2. This Broad Biodistribution Should Have Made the Carrying Out of Controls Required for GTP Essential

##### Germline Integration

The possibility of a vertical germline needs to be investigated (signals in gonads, signals in gametes, semen fractionation studies and integration analysis), especially since the EMA emphasizes a broader biodistribution pattern with low and measurable radioactivity in the ovaries and testes [11]. Genome integration studies are required for GTMP [24]. It was specified in 2009 that, for gene therapy medicinal products not expected to be capable of integration, integration studies must be carried out if the biodistribution of the product indicates a risk of germline transmission [13]. A 2005 document, dedicated specifically to the study of germline transmission of gene transfer vectors and naked DNA, specifies that only DNA, and not RNA, is presumed to pose a risk of germline modification [60]. This assertion can be questioned on the basis of two publications showing that, firstly, SARS-CoV-2 RNA can integrate into the genome [61] and, secondly, that the vaccine mRNA may be able to integrate into the genome of human cells in culture [62]. Although contested [63], these results would tend to require genome integration studies for mRNAs, especially since spike mRNA also translocates into the nucleus [64].

##### Genotoxicity

Preclinical genotoxicity studies in rats for Moderna showed equivoqual results [65] (p. 21). The conclusion was that “Overall, the genotoxic risk to humans is considered to be low due to minimal systemic exposure following IM administration, limited duration of exposure, and negative in vitro results”. It would have been wise to continue these studies, since exposure is not limited to muscle, nor is the duration of exposure. As specified in the EMA regulations for GTPs “If a positive finding occurs, additional testing will be needed to ensure the safety of the product” [38].

##### Reproductive and Developmental Toxicity 

According to a document concerning non-clinical trials of the Moderna vaccine obtained by the FOIA, skeletal variations (one or more rib nodules and one or more wavy ribs with no effect on the viability or growth and development of F1 generation) appeared in the fetuses of vaccinated rats but were not considered adverse. However, it is emphasized that they appeared at the same time as maternal toxicity correlating with the most sensitive period for rib development in rats [65].

According to documents obtained by the FOIA from Australian [66] and Japanese [67] regulatory agencies, skeletal malformations were also found in the Pfizer preclinical trial. The incidence of supernumerary lumbar ribs was higher in the treatment group compared with the control group, but was not considered to be treatment-related. This concordance of fetal anomalies with the two types of mRNA vaccines should have led to more detailed studies.

##### Pharmacokinetics of Nanomedicine According to the FDA

For a new excipient, the chemical, pharmaceutical and biological information should be identical to that provided for the active substance [17] (p. 135). Furthermore, according to Hemmrich and McNeil [68], the status of LNP components is confused, according to the FDA; they are considered either as “starting materials” (and therefore not as excipients) or as “inactive ingredients” (and therefore excipients) according to the documents whereas, according to the FDA itself, they should be considered as active ingredients. According to the same authors, developers must demonstrate the safety of the new ingredient, and “excipients intended for long-term use may require repeated dose toxicology studies over 6 months and carcinogenicity studies over 2 years”. Indeed, the manufacturers intended only two doses of vaccine [46], but some populations are currently receiving up to six doses spaced a few months apart. The stability, toxicity and biodistribution of the intact nanoparticle containing the mRNA, the active substance, must be evaluated, rather than the isolated lipid components, contrary to what has been carried out (Moderna and Pfizer have partially evaluated the biodistribution of lipids in nanoparticles, or of nanoparticles containing mRNAs other than those used in anti-COVID-19 vaccines) [11,32,33,34].

The FDA classification of these mRNAs as GTPs would have resolved these ambiguities since the FDA recommends assessing the risks of the GTP delivery procedure (biodistribution in blood, cerebrospinal fluid, germline, heart and brain to be assessed in preclinical trials, persistence of the vector). The FDA also requires the evaluation of potential horizontal transmission of the replication-competent vectors from the patient to family members and health care providers (i.e., shedding). This requirement should have applied even if the vector is not a replication-competent virus.

### 4.4. Clinical Studies

As shown above, the spike protein has been shown to circulate well in the blood, excretion studies should therefore have been carried out. The CHMP noted that no data are available on vaccine placental transfer or excretion in milk [11] (p. 56). Studies independent of the manufacturers have shown the passage of vaccine mRNA into breast milk in the first week following injection [69,70,71,72] and adverse effects on breast-fed babies could be due to this passage, according a FDA report [73]. Nanoparticles, similar to those in COVID-19 mRNA vaccines, have been shown to be able to cross the placental barrier in mice [74]. Extensive preclinical and clinical studies should have explored this passage in milk and through the placenta.

Carcinogenicity, tumorigenicity and immune suppression studies should have been carried out, because two studies have suggested that mRNA vaccines may induce immunotolerance [75,76]. In addition, the spike protein may interact with the tumor suppressor [77,78] (p. 53). It would therefore be wise to explore the tumorigenic effect in vivo and to monitor any cancers developed by vaccinated individuals over the long term, especially as it has been suggested that cancers can be reactivated by mRNA vaccines [79,80,81] or may develop after mRNA vaccination [82,83,84,85,86].

### 4.5. Vaccinovigilance

GTP regulations require the very long-term monitoring of adverse effects. This will be difficult to achieve for mRNA vaccines because the EMA has requested a 24-month follow-up of adverse events after vaccination, pointing out that a significant number of participants in the placebo group were vaccinated, which makes this follow-up more difficult [34] (pp. 14, 114 and 138). Moderna announced that “as of 13 April 2021 all placebo participants have been offered the Moderna COVID-19 vaccine and 98% of those have received the vaccine” [87].

The latest date for pharmacovigilance follow-up required by the EMA is 31 March 2024 [11], well below the FDA’s long-term follow-up of adverse events for GTP, which is 5 to 15 years, and 30 years for the EMA. In addition to the extensive pharmacovigilance plan requested by the EMA [11], we could call for reinforced monitoring. As the placebo groups in the clinical trials were vaccinated, the long-term monitoring of these adverse effects could be carried out by retrospective observational studies comparing the incidence of pathologies according to participants’ vaccination status. Two publications about adenovirus vectors [88,89] suggest that the long-term effects of gene therapy vectors, although they are not specifically mRNA vectors, clearly exemplify the lack of study and potential risks in the long term.

The FDA and EMA recommend the long-term monitoring of possible adverse effects of GTPs, particularly for certain diseases (cancers, hematological, neurological, rheumatological conditions and infections). We have seen above that there are reported cases of new or reactivated cancers following anti-COVID-19 mRNA vaccination. There are also reports of cases of diseases specifically to be monitored after GTP administration. Here are just a few examples, as it is impossible to be exhaustive. Concerning hematological disorders, there have been reports of bone marrow suppression [90] and aplastic anemia [91]. For neurological conditions, encephalitis [57], rhomboencephalitis [92], demyelinization [93] and autoimmune neurological diseases [94] were noted. Rheumatological conditions [95], de novo autoimmune rheumatic diseases [96], autoimmune like myopathy [97] and new or exacerbated inflammatory diseases [94] have also been reported.

A recrudescence of brain abscesses in 2021, after the start of the massive vaccination campaign, is reported [98] and non-covidial pneumonias following mRNA vaccines [99].

The EMA points out that an interventional study is underway to assess the safety and tolerability of Pfizer’s vaccine in pregnant women [11]; although the actual study completion date is 15 July 2022, no results have been published [100], which is unfortunate.

## 5. Conclusions

Although the principle of action of COVID-19 mRNA vaccines corresponds to the definition of gene therapy products (GTPs), they have been excluded from the regulation of GTPs by the regulatory agencies (US-FDA and EMA) and subjected to the regulation of vaccines against infectious diseases. No scientific or ethical justification is given for this exclusion, and there remain inconsistencies in the regulations. For example, under European and French regulations, a vaccine must contain an antigen, which is not the case for mRNA vaccines. These products could be considered “pro-vaccine”. In fact, mRNA vaccines do not contain an antigen, but make the vaccinee produce it. They can therefore be classed as pro-drugs or “pro-vaccine”. Special regulations should be drawn up for this type of product, insisting on potency controls, i.e., the quality, quantity, duration and sites of expression of the antigen of interest, as well as the toxicity of this antigen. As proposed at the start of 2020, the SARS-CoV-2 spike protein interacts with the renin-angiotensin system [101,102,103] and has a recognized toxicity that was known since before COVID-19 [104] and has been confirmed since [105,106,107,108].

According to European regulations, vaccines are human medicinal products and must therefore undergo the same controls, but not all of these controls are generally applied to vaccines against infectious diseases. With regard to the controls applied to mRNAs, it is worth noting that the degree of purity of the product is lower than that required for any drug: this is questionable for a new formulation and principle of action. It is also possible that batch heterogeneity was not detected by the batch release procedure. Impurities linked to this new formulation could pose safety problems; the presence and quantity of contaminating DNA from the template used to manufacture the RNA and of ds-RNA would need to be reassessed. The presence of antibiotic resistance genes in contaminating template DNA also raises safety issues.

Pharmacokinetic studies are not generally required for vaccines, except in the case of new formulations, which is the case here. However, extensive studies in this field would have been necessary, since they did not detect the wide distribution and persistence of mRNA, and its product, the spike protein, in the bodies of vaccinees, the passage of mRNA in breast milk, nor the possible passage through the placenta of vaccinated mothers. GTP regulations require these in-depth studies on the complete formulation (the lipid nanoparticle loaded with the mRNA corresponding to the drug product).

Because of this wide and persistent biodistribution, essential tests required for GTPs should have been carried out regarding: the risk of genotoxicity, genome integration and germ-line transmission, insertional mutagenesis, tumorigenicity, embryo/fetal and perinatal toxicity, long term expression, repeated toxicity and excretion in the environment (shedding in the seminal fluid, for example).

The long-term safety monitoring of GTPs is required over several years whereas, for vaccines, it is generally only carried out over a few weeks. This should not be acceptable, given the persistence of the drug product and the expressed protein. The known results of anti-cancer therapies and mRNA vaccines could lead us to anticipate problems of safety and efficacy. In the case of anti-cancer mRNAs, the vast majority of open-label clinical trials have been carried out on very small numbers of patients, with either unpublished or negative results [109,110]. Randomized studies also showed negative results, reporting more frequent adverse events in the treatment group [111,112]. Concerning infectious diseases, two trials of mRNA vaccines encapsulated in LNPs showed notable adverse effects. A trial of an mRNA vaccine against rabies showed numerous adverse effects superior to those of the classic vaccine, which is already very reactogenic, notably lymphopenia (this effect was also found for anti-COVID-19 mRNA vaccines) [113]. An influenza vaccine trial [114] showed severe adverse effects in humans (31 subjects were observed over only 43 days and at least 4 serious adverse effects were found). In a non-randomized trial against HIV [115], the response was inexplicably incomplete in some patients. According to another HIV trial of 15 participants against a placebo, immune responses were unsatisfactory and of limited duration [116]. The founder of BioNTech himself, Ugur Sahin, warned against the use of codon optimization, which can alter translation speed and lead to misfolding. He also underlined the potential toxicity of unnatural nucleotides. He also mentioned the wide biodistribution of mRNA injected intramuscularly. He reminded us that we should fear the appearance of anti-self mRNA antibodies in patients suffering from autoimmune diseases [27].

The role of regulatory agencies is to ensure the safety and efficacy of medicines. The COVID-19 pandemic emergency has accelerated the timetable for the production and clinical use of COVID-19 vaccines; it is, therefore, possible that certain safety aspects have not been fully addressed. It is, therefore, important to take these aspects into account in the future, so as not to undermine public confidence in vaccines in general.

The WHO declared an end to the emergency phase of the COVID-19 pandemic at the beginning of May 2023 but will continue to authorize the use of the Emergency Use Listed (EUL) procedure. The emergency authorization of vaccines should be transformed into prequalification via a smooth transition [117]. However, a wide-ranging public discussion should be opened on this transition to the routine use of mRNA vaccines, without them being subject to the controls required for GTPs.

In the EMA document designed to regulate the clinical evaluation of new vaccines from 2023, there is no mention of mRNA vaccines, and it is still specified that vaccines contain antigens; this document would therefore not apply to mRNA vaccines that do not contain antigens. It is once again specified that nonclinical pharmacokinetic studies might be applicable when new delivery systems are employed or when the vaccine contains novel adjuvants or excipients. It is a pity that these points have not been specified specifically for mRNA vaccines [118]. An article from early 2021 [119] emphasized the need for further studies to ensure the quality, efficacy and safety of mRNA vaccines; it was written before these products were marketed. It seems important to clarify which additional controls should be required in light of the detailed results of the preclinical trials and safety data published in the post-marketing phase.

In the future, it should be discussed whether all mRNA-based products should be subject to the same regulations and controls, whether or not they are considered vaccines. It is not justifiable to subject therapeutic mRNAs to strict controls when they are intended for patients representing a small proportion of the human population, and to exclude from these controls mRNA vaccines intended for the majority of the healthy human population.

## Figures and Tables

**Table 1 ijms-24-10514-t001:** History of gene therapy vs. vaccine regulation.

Year	Regulatory Agency	Rule	Comment
1996	FDA [13] Points to Consider on Plasmid DNA Vaccines for Preventive Infectious Disease Indications	Plasmid DNA vaccines are defined as purified preparations of plasmid DNA, designed to contain a gene or genes for the intended vaccine antigen, as well as genes incorporated into the construct that allow for production in a suitable host system	No mention of RNA because RNA-based gene therapy was not yet envisaged
1998	FDA [12] content current as of 2021 Guidance for human cell therapy and gene therapy	Virus or DNA preparations used as preventive vaccines are not covered by this document, though there is some overlap in the issues	No mention of RNA because RNA-based gene therapy was not yet envisaged
2003	European Union Directive 2001/83/EC which regulates medicinal products for human use, amended in June 2003 Part IV relating to Advanced Medicinal Products (ATMPs) https://eur-lex.europa.eu/legal-content/EN/TXT/?uri=CELEX:32003L0063&qid=1686154511801, accessed on 8 June 2023	Gene Therapy Medicinal Products (GTMPs) are defined as: “a product obtained through a set of manufacturing processes aimed at the transfer, to be performed either in vivo or ex vivo, of a prophylactic, diagnostic or therapeutic gene (i.e., a piece of nucleic acid), to human/animal cells and its subsequent expression in vivo”	Specific GTMPs included “naked nucleic acid” This definition would include mRNA vaccines
2005	WHO [23]	WHO grant nucleic-acid-based vaccines the status of vaccines	Vaccines must comply with GMP In case of new formulations: distribution studies and toxicology studies for new additives are required
2007	FDA [20]	Manufacturing issues and preclinical required studies for DNA plasmids as vaccine to prevent infectious diseases	DNA plasmids are subject to controls inspired by those for GTP
2009	European Union Directive 2001/83/EC Part IV on ATMPs amended 14 September 2009 [13]	A GTP *(a) contains an active substance which contains or consists of a recombinant nucleic acid used in or administered to human beings with a view to regulating, repairing, replacing, adding or deleting a genetic sequence;* and *(b) its therapeutic, prophylactic or diagnostic effect relates directly to the recombinant nucleic acid sequence it contains, or to the product of genetic expression of this sequence.* *Gene therapy medicinal products shall not include vaccines against infectious diseases.*	Vaccines against infectious diseases are excluded from the GTP regulations No ethical or scientific justification is provided
2013	FDA [21]	Regulation of gene therapy products did not apply to vaccines against infectious diseases	Apply to DNA plasmids
2015	EMA [24] Reflection Paper on Classification of Advanced Therapy Medicinal Products	EMA confirms that vaccines against infectious diseases are not classified as gene therapy products	No ethical or scientific justification is provided
2016	EMA [25]	EMA specifies, in this document, that the non-clinical specific aspects of nucleic acid vaccines must be studied in light of GTP regulations	Does not include mRNAs but the definition provided is not exhaustive

## Data Availability

Not applicable.

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
