# Peer review of "mRNA: Vaccine or Gene Therapy? The Safety Regulatory Issues"

_ijms, 2023, doi:10.3390/ijms241310514_

Round 1

Reviewer 1 Report

The series of information reported in the text is proposed with a clear forcedly critical and often disparaging interpretation of the validity and use of RNA vaccines. Many of the observations regarding the production timing and clinical use of the anti-covid vaccine should consider the emergency events that led to the development of effective protection systems against the pandemic experienced in the last two years in a more analytical and less biased way.

The knowledge and study of the potential of RNA therapy is not recent, indeed it has been known for over 20 years, and this, in combination with the extensive collaboration and the commitment of funds, has meant the therapeutic success and the formalization of the end of the health emergency by the World Health Organization declaring its end after just over three years earlier, on 11 March 2020, with the declaration of the beginning of the pandemic.

Text needs to be reordered due to overlapping, unformatted sentences.

Tab.1 must be formatted and reset in schematic order without overlapping.

An accurate re-reading of the text is required due to the presence of formal errors in the language and the presence of many typing errors.

The bibliography needs to be reset and reordered, there are different fonts and numerous macro links to websites.

An accurate re-reading of the text is required due to the presence of formal errors in the language and the presence of many typing errors.

Author Response

Response to rewiever 1

Thank you for reviewing my text. Your comments have enabled me, I hope, to improve the manuscript. In fact, I probably didn't emphasize enough the pandemic emergency, which accelerated the evaluation schedule for COVID mRNA vaccines and explains why certain safety aspects were not fully addressed. I've added a few lines on this subject in the summary and conclusion.

With regard to knowledge acquired before the pandemic about mRNA products (vaccines against infectious diseases and anti-cancer drugs), I have also added a paragraph showing that certain efficacy and safety problems could be anticipated.

I have also carefully proofread the text and corrected any errors.

Here are the additions you will find in red in the revised manuscript

The role of regulatory agencies is to ensure the safety and efficacy of medicines. The COVID-19 pandemic emergency has accelerated the timetable for the production and clinical use of COVID vaccines: it is therefore possible that certain safety aspects have not been fully addressed. It is therefore important to take these aspects into account in the future, so as not to undermine public confidence in vaccines in general.

The known results of anti-cancer therapies and mRNA vaccines could lead us to anticipate problems of safety and efficacy. In the case of anti-cancer mRNAs, the vast majority of open-label clinical trials have been carried out on very small numbers of patients, with either unpublished or negative results.

 [Wei J, Hui AM. The paradigm shift in treatment from Covid-19 to oncology with mRNA vaccines. Cancer Treat Rev. 2022 Jun;107:102405. doi: 10.1016/j.ctrv.2022.102405]

[Lorentzen CL, Haanen JB, Met Ö, Svane IM. Clinical advances and ongoing trials on mRNA vaccines for cancer treatment. Lancet Oncol. 2022 Oct;23(10):e450-e458. doi: 10.1016/S1470-2045(22)00372-2]

Randomized studies also showing negative results report more frequent adverse events in the treatment group 

[Sebastian, M., Schröder, A., Scheel, B. et al. A phase I/IIa study of the mRNA-based cancer immunotherapy CV9201 in patients with stage IIIB/IV non-small cell lung cancer. Cancer Immunol Immunother 68, 799-812 (2019). https://doi.org/10.1007/s00262-019-02315-x]  

[A. Stenzl, S. Feyerabend, I. Syndikus, T. Sarosiek, H. Kübler, A. Heidenreich, R. Cathomas, C. Grüllich, Y. Loriot, S.L. Perez Gracia, S. Gillessen, U. Klinkhardt, A. Schröder, O. Schönborn-Kellenberger, V. Reus, S.D. Koch, H.S. Hong, T. Seibel, K. Fizazi, U. Gnad-Vogt,

1149P - Results of the randomized, placebo-controlled phase I/IIB trial of CV9104, an mRNA-based cancer immunotherapy, in patients with metastatic castration-resistant prostate cancer (mCRPC),

Annals of Oncology, Volume 28, Supplement 5,2017, https://doi.org/10.1093/annonc/mdx376.014]

Concerning infectious diseases, two trials of mRNA vaccines encapsulated in LNPs showed notable adverse effects. A trial of an mRNA vaccine against rabies showed numerous adverse effects superior to those of the classic vaccine, which is already very reactogenic, notably lymphopenia (this effect was also found for anti-Covid mRNA vaccines) [Aldrich C, Leroux-Roels I, Huang KB, et al. Proof-of-concept of a low-dose unmodified mRNA-based rabies vaccine formulated with lipid nanoparticles in human volunteers: A phase 1 trial. Vaccine. 2021 Feb;39(8):1310-1318. DOI:10.1016/j.vaccine.2020.12.070].

An influenza vaccine trial [Bahl K, Senn JJ, Yuzhakov O, Bulychev A, Brito LA, Hassett KJ, Laska ME, Smith M, Almarsson Ö, Thompson J, Ribeiro AM, Watson M, Zaks T, Ciaramella G. Preclinical and Clinical Demonstration of Immunogenicity by mRNA Vaccines against H10N8 and H7N9 Influenza Viruses. Mol Ther. 2017 Jun 7;25(6):1316-1327. doi: 10.1016/j.ymthe.2017.03.035] showed severe adverse effects in humans (31 subjects were observed over only 43 days and at least 4 serious adverse effects were found).

In a non-randomized trial against HIV [Van Gulck, Ellena,*; Vlieghe, Erikab,*; Vekemans, Marcb,*; Van Tendeloo, Viggo F.I.c,d,*; Van De Velde, Annc,d; Smits, Evelienc,d; Anguille, Sébastienc,d; Cools, Nathaliec,d; Goossens, Hermanc; Mertens, Liesbetb; De Haes, Winnia; Wong, Johnssone; Florence, Ericb,*; Vanham, Guidoa,f,g,*; Berneman, Zwi N.c,d,*. mRNA-based dendritic cell vaccination induces potent antiviral T-cell responses in HIV-1-infected patients. AIDS 26(4):p F1-F12, February 20, 2012. | DOI: 10.1097/QAD.0b013e32834f33e8]  the response is inexplicably incomplete in some patients. According to another HIV trial of 15 participants against placebo, immune responses are unsatisfactory and of limited duration [Gandhi RT, Kwon DS, Macklin EA, Shopis JR, McLean AP, McBrine N, Flynn T, Peter L, Sbrolla A, Kaufmann DE, Porichis F, Walker BD, Bhardwaj N, Barouch DH, Kavanagh DG. Immunization of HIV-1-Infected Persons With Autologous Dendritic Cells Transfected With mRNA Encoding HIV-1 Gag and Nef: Results of a Randomized, Placebo-Controlled Clinical Trial. J Acquir Immune Defic Syndr. 2016 Mar 1;71(3):246-53. doi: 10.1097/QAI.0000000000000852].

The founder of BioNTech himself, Ugur Sahin, warned against the use of codon optimization, which can alter translation speed and lead to misfolding. He recalled the potential toxicity of unnatural nucleotides. He also mentioned the wide biodistribution of mRNA injected intramuscularly. He reminded us that we should fear the appearance of anti-self mRNA antibodies in patients suffering from autoimmune diseases [27].

Reviewer 2 Report

It seems to me an excellent manuscript, which addresses a genuine concern of health personnel who are experts in gene therapy. The issue directly involves the millions of users vaccinated against COVID-19.

I only have minor comments

1.- Table 1 is not displayed properly, perhaps due to a format problem. Please check.

1.- The issue of safety and possible long-term effects from my point of view is one of the most relevant. In conclusion it is mentioned that: “Long-term safety monitoring of GTPs is required over several years, whereas for vaccines it is generally only carried out over a few weeks. This should not be acceptable, given the persistence of the drug product and the expressed protein”. At this point it is important to note that long-term effects with gene therapy products may persist or have consequences for the rest of the individual's life, even though the applied genetic material is no longer present or expressed. Similarly, adverse effects are not necessarily due to the therapeutic sequence, but may also be a consequence of the vector used. Studies with adenoviral gene therapy vectors have shown in preclinical studies adverse effects that affect the individual for the rest of his life, even if the vector is no longer present. In the case of mRNA vectors, there are not enough long-term studies. Even following the regulations for gene therapy, the long-term effects (for decades) of gene therapies are still not fully known. I suggest adding these two references that show or suggest long-term effects of gene therapy vectors, which, although they are not specifically mRNA vectors, are clear to exemplify the lack of study and potential risks in the long term.

• Adenovirus 5 produces obesity and adverse metabolic, morphological, and functional changes in the long term in animals fed a balanced diet or a high-fat diet: a study on hamsters: This study suggests pidemiological studies for chronic liver and cardiovascular diseases in gene therapy . DOI: 10.1007/s00705-018-04132-6

• Use of adenovirus type-5 vector vaccines in COVID-19: potential implications for metabolic health? DOI: 10.23736/S2724-6507.22.03797-6

Author Response

Response to reviewer 2

Thank you for reviewing my manuscript.

I've sent the editor the raw file of Table 1, so that it can be properly included in the text.

I have taken your comment into account and included the publications mentioned: you will see this addition in red just before the conclusion.

Round 2

Reviewer 1 Report

considering the corrections made by the author, the new version of Ms-ijms-2470752 can be accepted.